# MBNL1-Associated Mitochondrial Dysfunction and Apoptosis in C2C12 Myotubes and Mouse Skeletal Muscle

**DOI:** 10.3390/ijms21176376

**Published:** 2020-09-02

**Authors:** Shingo Yokoyama, Yoshitaka Ohno, Tatsuro Egawa, Kazuya Ohashi, Rika Ito, Huascar Pedro Ortuste Quiroga, Tomohiro Yamashita, Katsumasa Goto

**Affiliations:** 1Laboratory of Physiology, School of Health Science, Toyohashi SOZO University, Toyohashi 440-8511, Japan; s-yokoyama@sozo.ac.jp (S.Y.); k-ohashi@sozo.ac.jp (K.O.); 2Faculty of Rehabilitation and Care, Seijoh University, Tokai 476-8588, Japan; yohno@sozo.ac.jp; 3Department of Physiology, Graduate School of Health Science, Toyohashi SOZO University, Toyohashi 440-8511, Japan; egawa.tatsuro.4u@kyoto-u.ac.jp (T.E.); s1355102@sc.sozo.ac.jp (R.I.); s1855105@sc.sozo.ac.jp (H.P.O.Q.); s1855104@sc.sozo.ac.jp (T.Y.); 4Laboratory of Sports and Exercise Medicine, Graduate School of Human and Environmental Studies, Kyoto University, Kyoto 606-8511, Japan

**Keywords:** MBNL1, PGC-1α, mitochondrial membrane potential, apoptosis, Bax

## Abstract

We explored the interrelationship between a tissue-specific alternative splicing factor muscleblind-like 1 (MBNL1) and peroxisome proliferator-activated receptor-γ coactivator 1-α (PGC-1α), B-cell lymphoma 2 (Bcl-2) or Bcl-2-associated X protein (Bax) in C2C12 myotubes and mouse skeletal muscle to investigate a possible physiological role of MBNL1 in mitochondrial-associated apoptosis of skeletal muscle. Expression level of PGC-1α and mitochondrial membrane potential evaluated by the fluorescence ratio of JC-1 aggregate to monomer in C2C12 myotubes were suppressed by knockdown of MBNL1. Conversely, the ratio of Bax to Bcl-2 as well as the apoptotic index in C2C12 myotubes was increased by MBNL1 knockdown. In plantaris muscle, on the other hand, not only the minimum muscle fiber diameter but also the expression level of MBNL1 and PGC-1α in of 100-week-old mice were significantly lower than that of 10-week-old mice. Furthermore, the ratio of Bax to Bcl-2 in mouse plantaris muscle was increased by aging. These results suggest that MBNL1 may play a key role in aging-associated muscle atrophy accompanied with mitochondrial dysfunction and apoptosis via mediating PGC-1α expression in skeletal muscle.

## 1. Introduction

Skeletal muscle atrophy, which is defined as a loss of muscle mass, is induced by inactivity [1,2,3], disease [4,5,6], or aging [7,8,9]. Aging-associated muscle atrophy (so-called sarcopenia), is accompanied by muscle weakness, resulting in impaired muscle function including reduced force generation [10] and aerobic capacity [8]. Several hypotheses have been proposed as a cause of age-associated skeletal muscle atrophy, such as proteostasis disruption [11], mitochondrial dysfunction [12], and apoptosis [13]. Activation of apoptosis as well as the upregulation of mitochondrial apoptosis-regulatory proteins including B-cell lymphoma 2 (Bcl-2) and Bcl-2-associated X protein (Bax) has been reported in aged skeletal muscle [14]. However, the molecular mechanisms involved in aging-associated mitochondrial apoptosis have not been fully elucidated.

Several characteristics of sarcopenia are consistent with the muscular symptoms of myotonic dystrophy type 1 (DM1), including insulin resistance, satellite cell senescence, and mitochondrial dysfunction [15]. DM1 is caused by a CTG trinucleotide expansion in the 3′ untranslated regions (UTR) of DM1 protein kinase (DMPK) gene on chromosome 19 [16,17,18]. Although the molecular mechanisms by which this expanded repeat produces the pathophysiology of DM1 are not unraveled, muscleblind-like 1 (MBNL1), an RNA-binding protein, is proposed as a key molecule in DM1 pathogenesis [19]. In fact, MBNL1-null mice result in skeletal muscle myotonia and histopathology that are characteristic of DM1 [20]. Since MBNL1 is a tissue-specific regulator of developmentally programmed alternative splicing [21], MBNL1 may play a key role in the development of age-associated muscle weakness.

Mitochondrial dysfunction is postulated to be linked with aging-associated impairment of skeletal muscle [22]. Peroxisome proliferator-activated receptor-γ coactivator 1-α (PGC-1α) is a principal regulator of mammalian mitochondrial biogenesis [23]. Expression level of PGC-1α in skeletal muscle decreases with aging [13], and the upregulation of PGC-1α in mouse skeletal muscle attenuates aging-associated mitochondrial dysfunction and sarcopenia [24].

PGC-1α also plays a role in the regulation of apoptosis. Absence of PGC-1α upregulates Bax expression and increases the ratio of Bax to Bcl-2 [25]. Mitochondrial-associated apoptosis is promoted by pro-apoptotic Bax and is suppressed by anti-apoptotic Bcl-2 [9,26,27,28]. In fact, it has been reported that the increase in ratio of Bax to Bcl-2 in skeletal muscle induces mitochondrial apoptosis [9,14,26]. Therefore, PGC-1α may be a key molecule in age-associated mitochondrial dysfunction as well as apoptosis. However, the molecular mechanism of aging-associated downregulation of PGC-1α in skeletal muscle remains unclear.

In the present study, we investigated the interrelationship between MBNL1 and PGC-1α, Bax, Bcl-2, or mitochondrial membrane potential in C2C12 myotubes and mouse skeletal muscle. We also explored the putative of MBNL1 in mitochondrial-associated apoptosis in skeletal muscle.

## 2. Results

### 2.1. Cell Culture Experiments

#### 2.1.1. Effect of MBNL1 Knockdown on Differentiation in C2C12 Cells

Firstly, we investigated the effects of MBNL1 knockdown on differentiation in C2C12 cells. Both mRNA and protein expression level of MBNL1 in C2C12 myotubes was decreased by knockdown of MBNL1 (siMBNL1). There was a significant difference in mRNA and protein expression level of MBNL1 between targeting and scrambled small interfering RNA (siRNA) for MBNL1 (−70% in mRNA: *p* < 0.05, −76% in protein: *p* < 0.05; Figure 1A,B,D). On the other hand, the effects of knockdown of MBNL1 had no impact on mRNA expression level of MBNL2, which is an isoform of MBNL (Figure 1C). The knockdown of MBNL1 showed a decreased content of muscle protein (−11%: *p* < 0.05; Figure 1E). However this did not affect the mRNA expression level of either myogenin or Creatine kinase (CK) protein, these molecules are generally accepted as a differentiation marker [29] (Figure 1A,C,F,G). Figure 2 shows the effects of MBNL1 knockdown on the morphology of C2C12 myotubes. MBNL1 knockdown also had no effect on myotube diameter and number of nuclei in a myotube.

#### 2.1.2. Effects of MBNL1 Knockdown on Gene Expression in C2C12 Myotubes

Next, we examined the genes fluctuated by MBNL1 knockdown in C2C12 myotubes using RNA sequencing (RNA-Seq). According to the cut-off point (false discovery rate (FDR) < 0.05 and |log2FC| > 1), 81 differentially expressed genes (DEGs) were identified between targeting and scrambled siRNA for MBNL1. Of these genes, 54 were upregulated and 27 were downregulated. DEGs associated with the membrane or plasma membrane were affected by MBNL1 knockdown (Figure 3). Table 1 shows the rate of variability of MBNL1 knockdown-associated downregulated genes. Interestingly, the expression level of *Bpifc*, *Ifitm6*, *Wwc1*, *Cyp11a1*, and *Ppargc1a* decreased by over 75%, compared to scrambled siRNA controls.

#### 2.1.3. Effect of MBNL1 Knockdown on the Expression Level of PGC-1α, Bax, and Bcl-2 in C2C12 Myotubes

Among the identified downregulated genes associated with downregulation of MBNL1, we focused on the *Ppargc1a* gene, which encodes the PGC-1α protein. PGC-1α is considered to be a master regulator of mitochondrial biogenesis [30]. RT-PCR analysis revealed that MBNL1 knockdown caused ~68% of decrease in the mRNA expression level of PGC-1α compared with scrambled siRNA (*p* < 0.05, Figure 4B). These findings not only validate the RNA-Seq data but are further substantiated at the protein level. Indeed, Western blot analysis showed that the expression level of PGC-1α was decreased by 23% following MBNL1 knockdown (*p* < 0.05, Figure 4A,C). Another interesting result was the increased expression level exhibited by Bax (pro-apoptotic protein), following MBNL1 knockdown (*p* < 0.05, Figure 4A,D). On the other hand, the expression level of the Bcl-2 (an anti-apoptotic protein) was also examined and found to be stably expressed following MBNL1 knockdown (Figure 4A,E). From these findings, one is able to calculate a Bax/Bcl-2 ratio, which serves as an index for mitochondria-mediated apoptosis. When the ratio was applied, we found that the knockdown of MBNL1 significantly increased mitochondria-mediated apoptosis compared to control conditions (*p* < 0.05, Figure 4F).

#### 2.1.4. Effect of MBNL1 Knockdown on Mitochondrial Membrane Potential and Apoptosis in C2C12 Myotubes

In the present study we evaluated mitochondrial function in C2C12-derived myotubes by analyzing the changes in the potential of the mitochondrial membrane. This was achieved using the fluorescent probe JC-1 (Figure 5A).

In healthy cells with high ΔΨM (important parameter of mitochondrial function), JC-1 forms complexes known as JC-1 aggregates with intense red fluorescence. Conversely, in cells with low ΔΨM, JC-1 remains in the monomeric form, which exhibits green fluorescence. Consequently, mitochondrial depolarization (a process that occurs in apoptosis [31]) is indicated by a decrease in the red/green fluorescence intensity ratio. Analysis of the data revealed that the fluorescence ratio of JC-1 was significantly decreased following the knockdown of MBNL1 (*p* < 0.05, Figure 5A,B). In order to substantiate these findings, we also examined the apoptotic index by counterstaining nuclei with Hoechst 33342 (Figure 5C). This enabled us to calculate a ratio between apoptotic nuclei (bright fluorescent chromatin, which is highly condensed or fragmented) to normal nuclei (blue chromatin with organized structure). Quantification of this data showed an increased apoptotic index in MBNL1-downregulated conditions compared to controls (*p* < 0.05, Figure 5D).

### 2.2. Animal Experiments

#### 2.2.1. Wet Weight and Fiber Size of Plantaris Muscles of Young and Old Mice

We next turned our attention to animal studies to deepen our understanding of MBNL1. Initially, a comparison of young versus old mice was made. Body weight, absolute muscle wet weight, and muscle weight relative to body weight in young and old mice are shown in Figure 6A–C. The body weight of old mice was significantly higher than that of young mice (*p* < 0.05, Figure 6A). However, the absolute muscle wet weight of plantaris muscles showed no significant change between young and old mice (Figure 6B). In addition, when the muscle weight relative to body weight was examined (Figure 6C), we found a lower muscle weight in aged mice compared to their young counterparts (*p* < 0.05). Figure 6D shows typical images from hematoxylin-eosin-stained plantaris muscles of young and old mice. The fiber size of plantaris muscle in old mice was significantly smaller than that of young mice (*p* < 0.05, Figure 6E,F).

#### 2.2.2. Expression Level of MBNL1, PGC-1α, and Apoptosis Regulatory Proteins in Plantaris Muscles of Young and Old Mices

To investigate a physiological role of MBNL1 in aging-associated muscle atrophy, we compared the expression level of MBNL1, PGC-1, Bax, and Bcl-2 in plantaris muscles of young and old mice. Expression levels of MBNL1 mRNA and protein in plantaris muscles of old mice were significantly lower than that of young mice (*p* < 0.05, Figure 7A–C). Although there was no significant difference at the mRNA level for PGC-1α (Figure 7D), at the protein level, muscle isolated from old mice showed a significantly lower expression of PGC-1α compared to that of young mice (*p* < 0.05, Figure 7A,E). The expression level of Bax protein in plantaris muscle of old mice was found to be significantly higher than that of young mice (*p* < 0.05, Figure 7A,F). Since there was no significant difference in the expression level of Bcl-2 protein in plantaris muscles between young and old mice (Figure 7A,G), the Bax/Bcl-2 ratio was increased by aging (*p* < 0.05, Figure 7H).

## 3. Discussion

The present study showed that MBNL1 knockdown depressed both mRNA and protein expression levels of PGC-1α. MBNL1 knockdown also showed a decrease in the fluorescence ratio obtained for JC-1 (aggregate to monomer in C2C12 myotubes) with the increased ratio of Bax to Bcl-2. Lower expression levels of MBNL1 (mRNA and protein) and PGC-1α (protein) and smaller muscle fiber size in plantaris muscles of old mice were observed compared to young mice. The ratio of Bax to Bcl-2 in mouse plantaris muscle was also increased by aging.

In this study, the expression levels of MBNL1 mRNA and protein in C2C12 myotubes were decreased by ~70% and ~76%, respectively, following siMBNL1 treatment. On the other hand, no effects of siMBNL1 treatment on mRNA expression level of MBNL2 was observed. However, MBNL1 knockdown seems to have no impact on the number of nuclei per myotube, myotube diameter, and the expression level of myogenin mRNA and CK protein—these molecules are generally accepted as a differentiation marker [29]. Therefore, MBNL1 has no role in myogenic differentiation. In this study, on the other hand, the total protein content in C2C12 myotubes was decreased by MBNL1 knockdown. In general, total protein content increases, accompanying myotube growth and differentiation [32,33,34]. Therefore, the decrease of total protein content induced by MBNL1 knockdown might indicate a decrease in the cytoplasmic protein contents in myotubes without any structural changes. However, we have no clear explanation regarding this issue at present.

In the present study, RNA-Seq analysis showed that the downregulation of MBNL1 yielded 81 DEGs, 54 of which were upregulated and 27 downregulated. Among the downregulated genes, we focused on the *Ppargc1a* gene, encoding the PGC-1α protein. The present study also showed that knockdown of MBNL1 is accompanied by a downregulation of PGC-1α at the mRNA level (~68% reduction) and the protein level (~23% reduction). There is no evidence that MBNL1 directly mediates the expression level of PGC-1α. Therefore, MBNL1 may play a role in the regulation of PGC-1α expression. Additional experiments are needed to elucidate this issue.

Since PGC-1α plays a key role in mitochondrial function and apoptosis in skeletal muscle [35], we investigated the effects of MBNL1 knockdown on mitochondrial membrane potential and apoptosis. MBNL1 knockdown suppressed the ratio of JC-1 aggregate to monomer fluorescence, indicating the loss of mitochondrial membrane potential. Since a decrease in mitochondrial membrane potential means mitochondrial dysfunction [36], MBNL1 has a potential role in maintaining mitochondrial function. In addition, the aforementioned decreased expression of PGC-1α following MBNL1 knockdown suggests a mechanism employed by MBNL1 in order to maintain mitochondrial integrity. To the best of our knowledge, this is the first report showing the physiological role of MBNL1 in the regulation of mitochondrial function.

It has been reported that the loss of total protein in C2C12 myotubes caused by H_2_O_2_-associated apoptosis [37]. Therefore, MBNL1 knockdown-associated decrease in total protein content in C2C12 myotubes might be attributed to apoptosis.

In the present study, the expression level of Bax, the ratio of Bax to Bcl-2, as well as the apoptotic index was increased by MBNL1 knockdown. It has been reported that the ratio of Bax to Bcl-2 is a valid indicator for developing apoptosis in various cells [38,39]. These results indicate that the reduced expression of MBNL1 induces apoptosis. Aging-associated increase Bax expression, and the ration of Bax to Bcl-2 was also observed in mouse skeletal muscle [25]. MBNL1 may have a protective role against mitochondrial-mediated apoptosis by regulating the expression of PGC-1α in skeletal muscle.

In this study, muscle wet weight relative to body weight, muscle fiber size, and the expression level of both MBNL1 and PGC-1α in plantaris muscle of old mice were lower than those of young mice. A previous study [40] showed higher expression of MBNL1 in old mice (28 months old) compared to the expression level of adult mice (9 months old). On the other hand, in the present study, MBNL1 expression level in young (2 months old) and old mice (22 months old) were investigated. Two-month-old young mice are in the developing stage of fiber size [41] and the number of satellite cells [42] in skeletal muscle, and there may be some different properties of skeletal muscle compared to 9-month-old adult mice. Further, in the previous report, the expression level of MBNL1 in quadriceps femoris muscles of *Balb-c* mice were investigated. In this study, on the other hand, MBNL1 expression level in plantaris muscle of *C57BL/6J* mice was investigated. Differences in mouse strains and/or muscle types may influence the expression level of MBNL1 with aging.

Conversely, the expression level of Bax protein and the ratio of Bax to Bcl-2 in plantaris muscle were significantly increased by aging. These results are consistent with the data obtained from our C2C12 experiments and by those previously reported [9,13,14,26]. Although the present study did not investigate the association between MBNL1 and PGC-1α in mouse plantaris muscle, the results from C2C12 experiment strongly suggest that aging-associated down-regulation of MBNL1 decreases the expression level of PGC-1α, then mitochondrial function is suppressed in aged skeletal muscle.

In conclusion, MBNL1 may play a key role in aging-associated muscle atrophy, a process accompanied with mitochondrial dysfunction and apoptosis.

## 4. Materials and Methods

### 4.1. Cell Culture Experiments

Cell culture experiments were carried using mouse myoblast-derived C2C12 cells following the methods previously described [43]. C2C12 cells were cultured on 6-well culture plates with type I collagen-coated surface (Corning Incorporated, Corning, NY, USA). Cells were grown in 2 mL of growth medium consisting of Dulbecco’s modified Eagle’s medium (DMEM, Thermo Fisher, Scientific, Waltham, MA, USA) supplemented with 10% heat-inactivated fetal bovine serum containing high glucose (4.5 g/L glucose, 4.0 mM L-glutamine without sodium pyruvate) for proliferation under a humidified atmosphere with 95% air and 5% CO_2_. At ~90% confluence, the medium was changed to the same amount of differentiation medium consisting of DMEM supplemented with 2% heat-inactivated horse serum containing low glucose (1.0 g/L glucose, 4.0 mM L-glutamine with 110 mg/L sodium pyruvate) to induce differentiation. The differentiation medium was changed every 2 days and the culture was maintained for 5 days.

Three days after the initiation of differentiation, we transfected RNA oligos into differentiating myotubes using Lipofectamine RNAiMAX Transfection Reagent (Thermo Fisher Scientific) as per the manufacturer’s instructions. Details for the transfection of siRNA were previously described [44]. Briefly, lipofectamine/siRNA complexes were added to the differentiation medium, and myotubes were incubated for 24 h. The final concentration of siRNA was set at 10 nM. Transfection medium was changed with freshly prepared differentiation medium following 24 h of the transfection. After 48 h from the transfection, myotubes were harvested for mitochondrial membrane potential assays, RNA-Seq, RT-PCR, or immunoblot analyses. Below are the siRNA oligonucleotides designed against mouse MBNL1 (siMBNL1: sense/antisense): 5′-GCAAUUUAGCAUGUUGGAATT-3′/5′-UUCCAACAUGCUAAAUUGCTT-3′. Cells in control cohorts were transfected with scrambled non-targeting control siRNA (siScramble), obtained from Takara Bio (Takara Bio, Otsu, Japan).

In this study, the number of the sample refers to the number of wells in each experimental condition. All analyses on each sample were carried out in duplicate.

### 4.2. Animal Experiments

All experimental procedures were carried out in accordance with the Guide for the Care and Use of Laboratory Animals as adopted and promulgated by the National Institutes of Health (Bethesda, MD, USA) and were approved by the Animal Use Committee of Toyohashi SOZO University (2007001). All treatments for animals were performed under anesthesia with intraperitoneal (*i.p.)* injection of sodium pentobarbital, and all efforts were made to prevent discomfort and suffering. Male *C57BL/6J* mice at 10 weeks (young; *n* = 5) and 100 weeks (old; *n* = 7) of age were used in this experiment. In hindlimb skeletal muscle of ~100-week-old *C57BL/6J* mice, apoptotic rate, which is the relative percentage of terminal deoxynucleotidyltransferase-mediated deoxyuridine triphosphate nick end labeling (TUNEL)-positive myonuclei per total myonuclei, might be ~10% [45]. Mice in each group were housed in same-sized cages. All mice were housed in a vivarium room with a 12h/12h light–dark cycle with temperature and humidity maintained at ~23 °C and ~50%, respectively. Solid food and water were provided ad libitum. Plantaris muscles from each group were dissected from left hindlimbs of young and old mice. Muscles were trimmed of excess fat and connective tissues, weighed, frozen in liquid nitrogen, and stored at −80 °C.

### 4.3. Immunoblot Analyses

The protein expression level of MBNL1, muscle-type creatine kinase (CK), PGC-1α, Bax, and Bcl-2 were evaluated by standard immunoblot analyses, as described previously [3,43]. The cells in each well were rinsed twice with 1 mL of ice-cold phosphate-buffered saline. The cells were then collected in 0.3ml of cell lysis reagent (CelLytic TM-M, Sigma-Aldrich, St. Louis, MO, USA) with 1% (*v*/*v*) Protease/Phosphatase Inhibitor Cocktail (#5872, Cell Signaling Technology Inc., Danvers, MA, USA). Proximal portions of the frozen left plantaris muscles were homogenized in an isolation buffer of tissue lysis reagent (CelLytic-MT, Sigma-Aldrich) with 1% (*v*/*v*) Protease/Phosphatase Inhibitor Cocktail (#5872, Cell Signaling Technology) with a glass homogenizer. The homogenates were centrifuged at 15,000× *g* (4 °C for 15 min), and the supernatant was collected. A part of the supernatant was solubilized in SDS sample buffer (30% *v*/*v* glycerol, 10% *v*/*v* 2-mercaptoethanol, 2.3% *w*/*v* SDS, 62.5 mM Tris-HCl, 0.05% *w*/*v* bromophenol blue, pH 6.8) at a concentration of 0.5 mg of protein ml-1 and was incubated at 95 °C for 5 min. SDS-PAGE was carried out on 10% polyacrylamide containing 0.5% SDS at a constant current of 20 mA for 120 min, as described previously [3]. Equal amounts of protein (5 µg) were loaded on each gel. Molecular weight markers (#161-0374, Bio-Rad, Hercules, CA, USA) were applied to both sides of 14 lanes as the internal controls for the transfer process and electrophoresis.

Following SDS-PAGE, proteins were transferred to polyvinylidene fluoride membrane (0.2 µm pore size, Bio-Rad) at a constant voltage of 100 V for 60 min at 4 °C. The membranes were stained by Ponceau S solution (Sigma-Aldrich) to show equal loading, and blocked for 1 h at room temperature in a blocking buffer: 5% (*w*/*v*) skim milk with 0.1% Tween 20 in Tris-buffered saline (TBS) at pH 7.5. The membranes were then incubated for 2 h with polyclonal antibodies for MBNL1 (ab45899, Abcam, Cambridge, UK), CK (ab198235, Abcam), PGC-1α (sc-13067, Santa Cruz Biotechnology, Dallas, TX, USA), Bax (#2772S, Cell Signaling Technology), and Bcl-2 (#3498S, Cell Signaling Technology), and then reacted with secondary antibodies (goat anti-rabbit immunoglobulin G (IgG) horseradish peroxidase-linked antibody, Cell Signaling Technology). After the final wash, protein bands were visualized by chemiluminescence (ECL Select Western blotting kit; GE Healthcare, Chicago, IL, USA), and signal density was measured by Light-Capture (AE-6971) using CS Analyzer version 2.08b (ATTO Corporation, Tokyo, Japan). Each sample was investigated in duplicate to at least ensure that results were not influenced by loading errors. The densities of Ponceau S staining were carried out as the internal controls for the transfer process and electrophoresis. Standard curves were constructed during the preliminary experiments to ensure linearity.

### 4.4. RT-PCR Analyses

RT-PCR analyses were carried out to access the mRNA expression level of MBNL1, myogenin, and PGC-1α in C2C12 myotubes and plantaris muscles, as described previously [1]. Briefly, total RNA was extracted from the C2C12 myotubes and distal portions of the muscles using the miRNeasy Mini kit (Qiagen, Hiden, Germany), according to the manufacturer’s protocol. The RNA sample was reverse-transcribed to complementary DNA (cDNA) using PrimeScript RT Master Mix (Takara Bio, Otsu, Japan). RT-PCR was then performed on the cDNA (Thermal Cycler Dice Real Time System IIMRQ, Takara Bio) using Takara SYBR Premix Ex Taq II (Takara Bio). The real-time cycle conditions were 95 °C for 30 s, followed by 40 cycles at 95 °C for 5 s and at 60 °C for 30 s for mRNA.

The relative fold change of expression was calculated by the comparative threshold cycle (CT) method using Takara Thermal Cycler Dice Real Time System Software Ver. 4.00 (Takara Bio). To normalize for the amount of total RNA present in each reaction, we used β-actin as internal standard. Primers were designed using the Takara Bio Perfect Real Time Support System (Takara Bio). The following primers were used: MBNL1, 5′-GGACAGCTTGTAGTTTGCCAGGA-3′ (forward) and 5′-GCAGATTTGGCCCAATGGAG-3′ (reverse); myogenin, 5′- CAGTGAATGCAACTCCCACAG-3′ (forward) and 5′- TGGACGTAAGGGAGTGCAGA-3′ (reverse); PGC-1α, 5′-TGATGTGAATGACTTGGATACAGACA-3′ (forward) and 5′-GCTCATTGTTGTACTGGTTGGATATG-3′ (reverse); β-actin, 5′-CATCCGTAAAGACCTCTATGCCAAC-3′ (forward) and 5′-ATGGAGCCACCGATCCACA-3′ (reverse).

### 4.5. RNA-Seq

Total RNA was extracted from C2C12 myotubes treated by MBNL1 siRNA or scrambled siRNA using the miRNeasy Mini kit. The preparation of the cDNA library and the RNA sequencing was performed by Bioengineering Lab (Atsugi, Japan). The cDNA originating from the RNA fragments were paired and sequenced using NextSeq 500 (Illumina, San Diego, CA, USA).

Tophat2 (ver. 2.1.0) was used to map the sequencing reads to the mouse genome. The sequencing reads were counted using featureCounts (ver. 1.5.0-p3). DESeq2 was used to perform the analysis of differentially expressed genes (DEG). The cutoffs of the DEG approach were chosen as false positive rate (FDR) < 0.05 and |log2 (fold change)| > 1. Gene Ontology (GO) enrichment analysis was performed for DEGs in the gene co-expression network with DAVID (Database for Annotation, Visualization and Integration Discovery, http://david.abcc.ncifcrf.gov/).

All of the RNA-Seq data obtained in this study were deposited in the DNA Data Bank of Japan (DDBJ) Sequence Read Archive (DRA), and they are accessible through DRA accession number DRA010689.

### 4.6. Morphological Analyses of Myotubes

Myotube diameter was evaluated following the methods previously described [43]. Briefly, six fields were chosen randomly, and 150 myotubes were measured using Image J. The average diameter per myotube was calculated as the mean of three short-axis measurements taken along the length of the myotube. To evaluate differentiation level of myotubes, we also investigated the number of nuclei per myotube [44,46].

### 4.7. Morphological Analyses of Plantaris Muscles

Frozen plantaris muscles were cut cross-sectionally into halves. Serial transverse cryosections (8 μm thick) of the midbelly region of the proximal side were sliced at −20 °C and mounted on glass slides. The slides were air-dried and stained by using hematoxylin–eosin. The images of muscle sections were incorporated into a personal computer by using a microscope (BZ-X710, KEYENCE, Osaka, Japan). Minimum diameter of individual fibers [47] in plantaris muscles were measured using Image J. At least 100 randomly selected myofibers were measured.

### 4.8. Mitochondrial Membrane Potential and Apoptotic Index

Mitochondrial membrane potential in C2C12 myotubes were assessed using JC-1 mitochondrial membrane potential assay kit (Cayman Chemical Company, Ann Arbor, MI, USA), according to the manufacturer’s instructions. Briefly, JC-1 solution (100 μL/mL culture medium) was added to the culture medium and was then mixed gently. The cells were then incubated in a CO_2_ incubator at 37 °C for 15 min. Images were then viewed and scanned by a fluorescence microscope (BZ-X710, KEYENCE, Osaka, Japan), at 470 nm excitation and 525 nm emission for green, and at 560 nm excitation and 630 nm emission for red. Red emission of the dye represented a potential-dependent aggregation of JC-1 in the mitochondria. Conversely, green fluorescence appearing in the cytosol after mitochondrial membrane depolarization represented the monomeric form of JC-1. More than 100 areas were selected from each group. The average intensity of red and green fluorescence was measured using Image J software (National Institutes of Health, Bethesda, MD, USA). The ratio of JC-1 aggregate (red) to monomer (green) intensity was then calculated.

To evaluate the contribution of apoptosis to MBNL1-associated changes in C2C12 differentiation, we stained cells with Hoechst 33342 to identify apoptotic nuclei. Briefly, a final concentration of 50 μg/mL Hoechst 33342 stain (Thermo Fisher Scientific, Waltham, MA, USA) was added to the culture medium. After 10 min of incubation, the medium was removed, and cells were rinsed in phosphate buffer solution. Analysis was performed immediately under fluorescence microscopy. The nuclei of healthy cells are generally spherical, and the DNA is evenly distributed. During apoptosis, the DNA becomes condensed, but this process does not occur during necrosis. Nuclear condensation can therefore be used to distinguish apoptotic cells from healthy cells or necrotic cells. Hoechst 33342 binds to DNA, and can be used to observe nuclear condensation [48]. To quantify the extent of apoptosis, we calculated an apoptotic index (the ration of condensed nuclei to normal nuclei) from six images of randomly selected fields of view for each treatment, as previously reported [49,50,51]. The apoptotic index is expressed as the proportion of apoptotic nuclei (brightly fluorescent, condensed compared to normal) relative to the total number of nuclei. These values are expressed as percentages.

### 4.9. Statistical Analyses

All values were expressed as means ± standard error of the mean (SEM). Statistical significance was analyzed using an unpaired Student’s *t*-test following the *F*-test for equal variances. The differences between groups were considered statistically significant at *p* < 0.05.

## Figures and Tables

**Figure 1 ijms-21-06376-f001:**
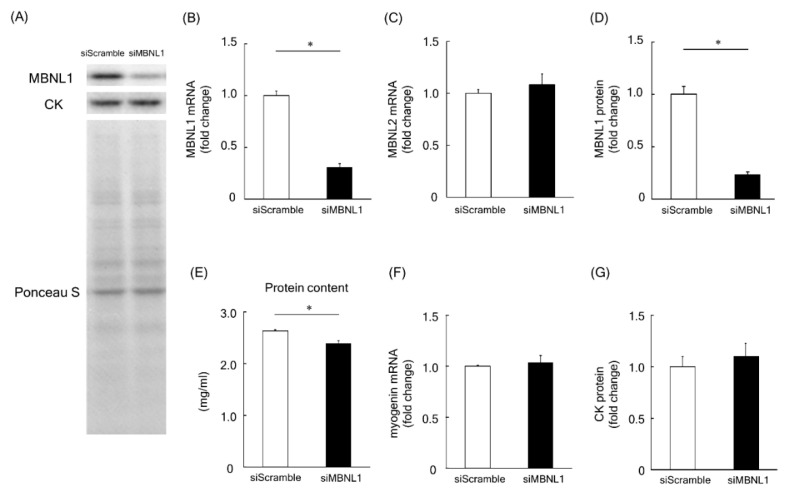
Effects of muscleblind-like 1 (MBNL1) knockdown on differentiation of C2C12 myotubes. (**A**) Representative expression patterns of MBNL1, Creatine kinase (CK), and loading control Ponceau S staining. mRNA expression level of MBNL1 (**B**) and MBNL2 (**C**). (**D**) Protein expression level of MBNL1. (**E**) Muscle protein content in 5 days after differentiation. The expression level of myogenin mRNA (**F**) and CK protein (**G**). siScramble: scrambled nontargeting small interfering RNA (siRNA), siMBNL1: targeting siRNA for MBNL1. Values are means ± standard error of the mean (SEM); *n* = 6 per group. *: *p* < 0.05.

**Figure 2 ijms-21-06376-f002:**
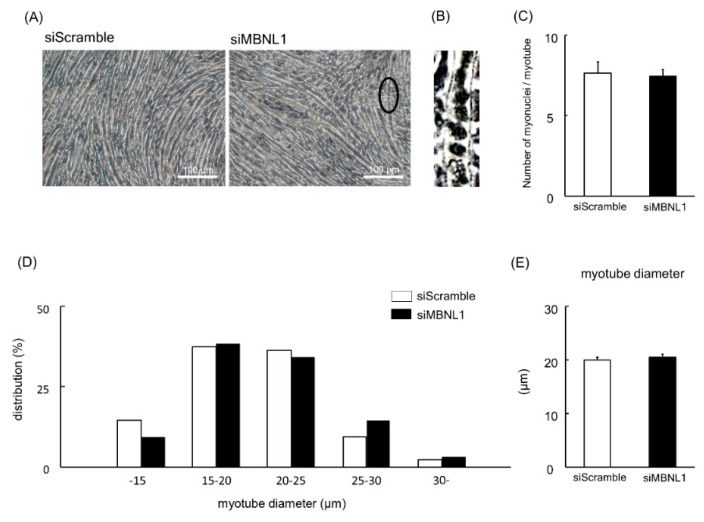
Effects of MBNL1 knockdown on the morphology of C2C12 myotubes. (**A**) Typical images of C2C12 myotubes. The ellipse in black in the image of MBNL1 indicates the enlarged area that is shown in panel B. (**B**) Enlarged image of the ellipse in black in the image (siMBNL1) in panel A. (**C**) The number of nuclei per myotube. (**D**) The frequency distributions of myotube diameter. (**E**) The average of myotube diameter. siScramble: scrambled nontargeting siRNA, siMBNL1: targeting siRNA for MBNL1. Values are means ± SEM; *n* = 6 per group.

**Figure 3 ijms-21-06376-f003:**
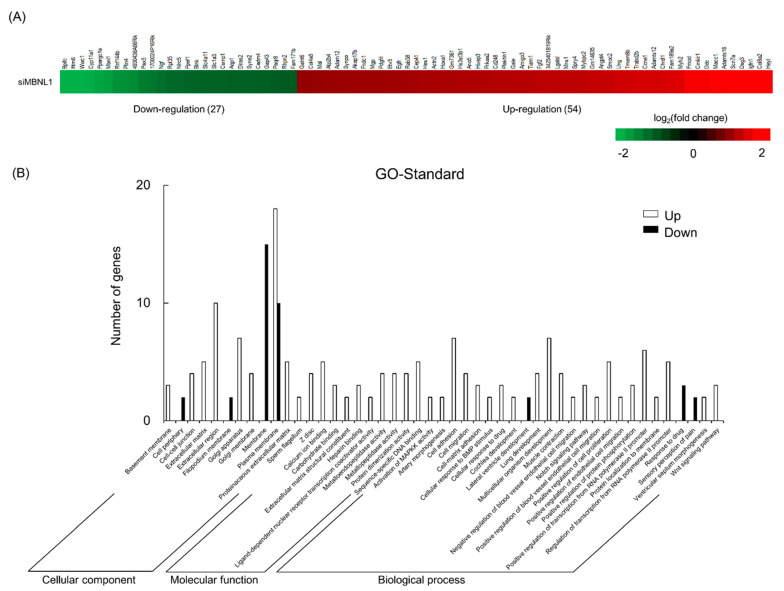
Effects of MBNL1 knockdown on the differentially expressed genes identification and Gene Ontology (GO) annotation in C2C12 myotubes. Heatmap (**A**) and GO annotation (**B**) of the differentially expressed genes (DEGs).

**Figure 4 ijms-21-06376-f004:**
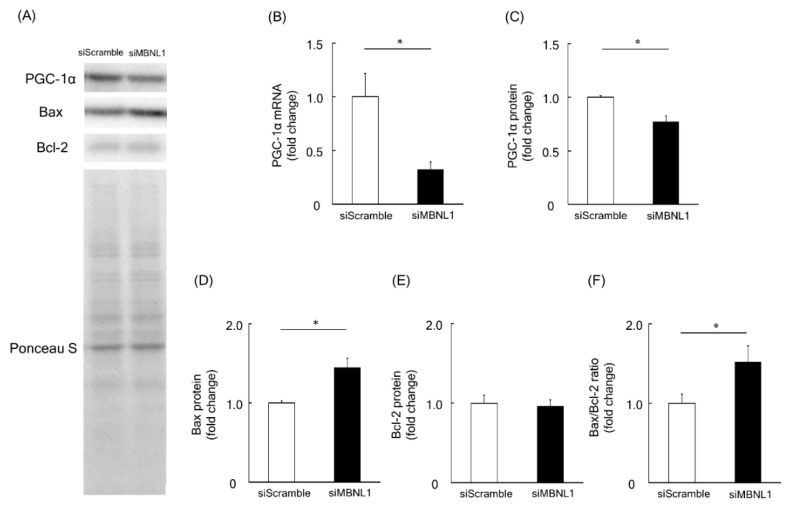
Effects of MBNL1 knockdown on the expression level of peroxisome proliferator-activated receptor-γ coactivator 1-α (PGC-1α), B-cell lymphoma 2 (Bcl-2) and Bcl-2-associated X protein (Bax) in C2C12 myotubes. (**A**) Representative expression patterns of PGC-1α, Bax, Bcl-2, and loading control Ponceau S staining. (**B**,**C**) mRNA and protein expression level of PGC-1α. The expression level of Bax and Bcl-2 proteins (**D**,**E**) and the ratio of Bax to Bcl-2 (**F**). Abbreviations are the same as in Figure 1. Values are means ± SEM; *n* = 6 per group. *: *p* < 0.05.

**Figure 5 ijms-21-06376-f005:**
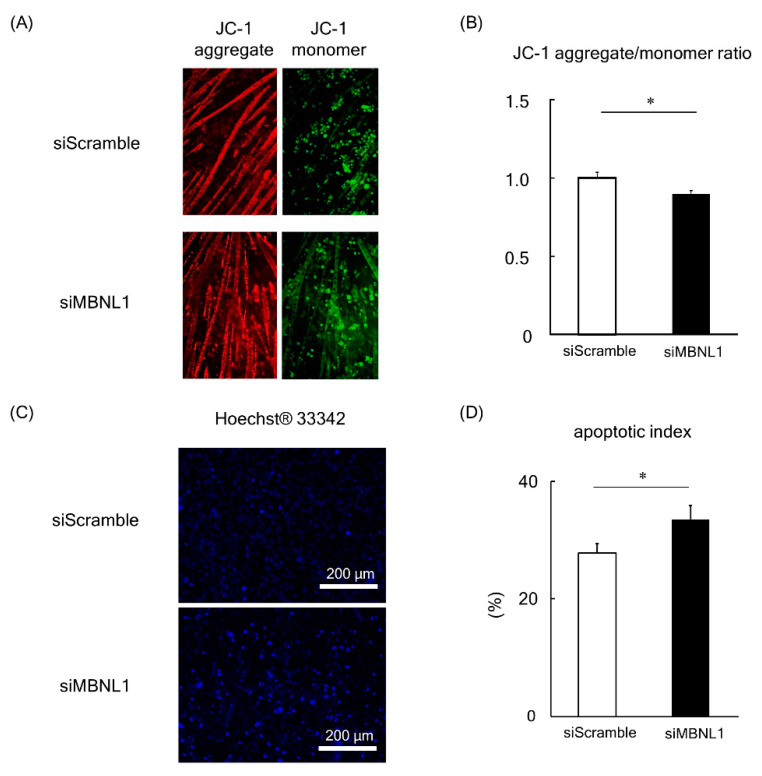
Effects of MBNL1 knockdown on mitochondrial membrane potential and apoptosis in C2C12 myotubes. Typical images of C2C12 myotubes stained by JC-1 (**A**) and Hoechst 33342 (**C**). (**B**) The red/green fluorescence intensity ratio of JC-1. (**D**) Apoptotic index, the ratio of apoptotic nuclei to normal nuclei. Abbreviations are the same as in Figure 1. Values are means ± SEM; *n* = 6 per group. *: *p* < 0.05.

**Figure 6 ijms-21-06376-f006:**
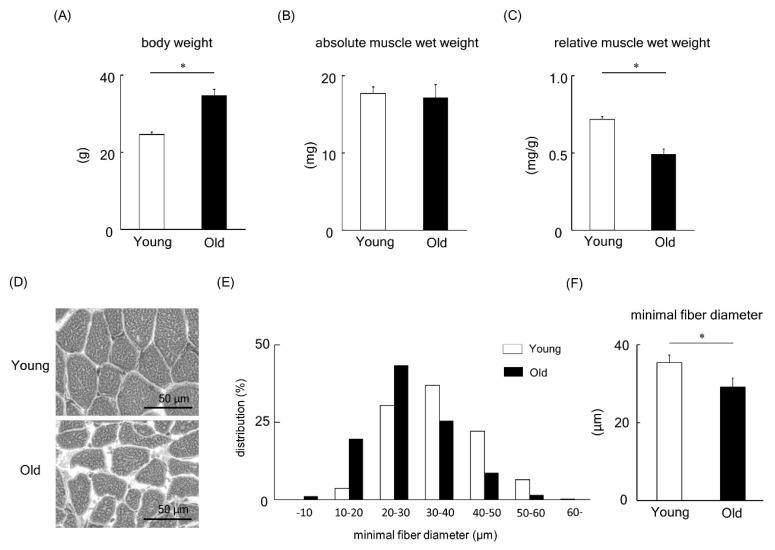
Body weight (**A**), absolute plantaris muscle wet weight (**B**), and muscle weight relative to body weight (**C**) in young and old mice. (**D**) Typical images from hematoxylin-eosin-stained plantaris muscles of young and old mice. The frequency distribution (**E**) and the mean value (**F**) of minimum muscle fiber diameter of plantaris. Values are means ± SEM; young: 10 weeks old, *n* = 5; old: 100 weeks old, *n* = 7. *: *p* < 0.05.

**Figure 7 ijms-21-06376-f007:**
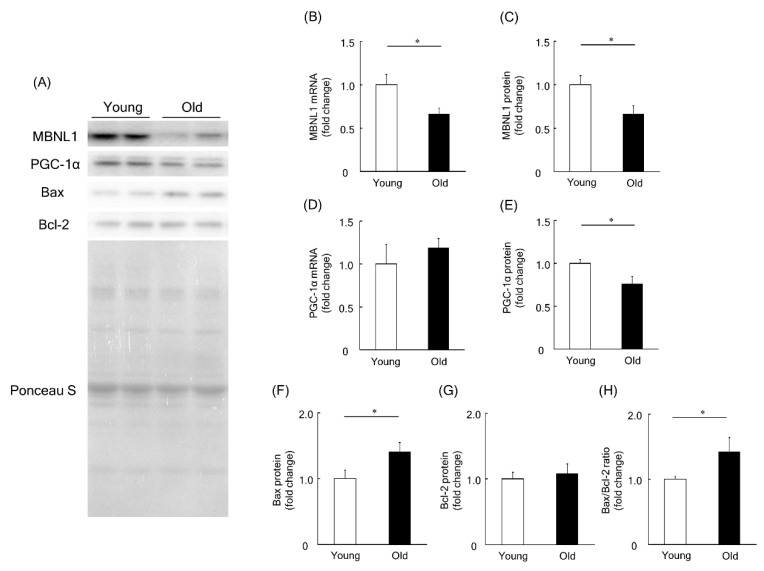
Expression level of MBNL1, PGC-1α, Bax, and Bcl-2 in plantaris muscles of young and old mice. (**A**) Representative expression patterns of MBNL1, PGC-1α, Bax, Bcl-2, and loading control Ponceau S staining. mRNA and protein expression level of MBNL1 (**B**,**C**) and PGC-1α (**D**,**E**). The expression level of Bax (**F**) and Bcl-2 (**G**) proteins and the ratio of Bax to Bcl-2 (**H**). Values are means ± SEM; young: 10 weeks old, *n* = 5; old: 100 weeks old, *n* = 7. *: *p* < 0.05.

**Table 1 ijms-21-06376-t001:** MBNL1 knockdown-associated downregulated genes with the rate of variability in C2C12 myotube.

ENSEMBL ID	Genes	log_2_ (Fold Change)	FDR Values
ENSMUSG00000050108	*Bpifc*	−2.64	0.01
ENSMUSG00000059108	*Ifitm6*	−2.49	0.03
ENSMUSG00000018849	*Wwc1*	−2.21	0.03
ENSMUSG00000032323	*Cyp11a1*	−1.95	0.03
ENSMUSG00000029167	*Ppargc1a*	−1.85	0.02
ENSMUSG00000027763	*Mbnl1*	−1.79	0.03
ENSMUSG00000038068	*Rnf144b*	−1.65	0.02
ENSMUSG00000002831	*Plin4*	−1.61	0.03
ENSMUSG00000069873	*4930438A08Rik*	−1.56	0.01
ENSMUSG00000014030	*Pax5*	−1.40	0.03
ENSMUSG00000078612	*1700024P16Rik*	−1.35	0.02
ENSMUSG00000027859	*Ngf*	−1.26	0.03
ENSMUSG00000062997	*Rpl35*	−1.20	0.05
ENSMUSG00000074151	*Nlrc5*	−1.18	0.01
ENSMUSG00000062168	*Ppef1*	−1.14	0.01
ENSMUSG00000061132	*Blnk*	−1.13	0.03
ENSMUSG00000074796	*Slc4a11*	−1.12	0.02
ENSMUSG00000005360	*Slc1a3*	−1.11	0.01
ENSMUSG00000032515	*Csrnp1*	−1.10	0.02
ENSMUSG00000004655	*Aqp1*	−1.09	0.01
ENSMUSG00000047842	*Diras2*	−1.09	0.04
ENSMUSG00000063450	*Syne2*	−1.08	0.01
ENSMUSG00000054793	*Cadm4*	−1.08	0.02
ENSMUSG00000047261	*Gap43*	−1.06	0.01
ENSMUSG00000025931	*Paqr8*	−1.05	0.02
ENSMUSG00000030494	*Rhpn2*	−1.02	0.04
ENSMUSG00000048388	*Fam171b*	−1.01	0.01

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
