# Peer review of "MBNL1-Associated Mitochondrial Dysfunction and Apoptosis in C2C12 Myotubes and Mouse Skeletal Muscle"

_ijms, 2020, doi:10.3390/ijms21176376_

Round 1

Reviewer 1 Report

Yokoyama et al. demonstrate correlative changes associated with MBNL1 knockdown in C2C12 cells. Overall the study offers very little insight into the function of MBNL1 in C2C12 cells.

Some concerns that need to be addressed are the following:

  1. A major limitation to the current study is that the authors have not validated their findings with a second siRNA targeting MBNL1.
  2. Specificity of the siRNA approach needs to be demonstrated, ie. Are the other MBNL isoforms also knocked down?
  3. Also, a dose dependent KD of MBNL1 would have strengthened the validity of the correlative changes reported.
  4. The data presented in Figure 4C are hardly convincing that a reduction of MBNL1 lead to greater apoptosis. Additional data are needed to support the authors claims.
  5. The results in Figure 6A contradict what has already been reported in the literature; that MBNL1 increases in skeletal muscle of aged mice relative to younger controls. See Malatesta et al. Eur J Histochem. 2013.

Reviewer 2 Report

Generally, the manuscript submitted by Yokoyama et al. is well-written, the experiments are well-designed, and the methodology is accurately described and well-performed. They explore the role of MBNL1 in mitochondrial membrane potential and apoptosis using C2C12 cell line RNA-seq and in vivo experiments. They show differences in the gene expression and protein levels of Mbnl1 in the planaris muscle of aged mice compared to young mice, and higher Bax/Bcl-2 protein ratio. They conclude that lower expression and protein levels of Mbnl1 during muscle ageing may be related to increased mitochondrial depolarization and apoptosis through higher Bax/Bcl-2 ratio; however, further analyses should be performed to corroborate the association of Pgdc-1alpha in vivo.  

The results are generally conclusive and of interest for those researchers involved in the muscle ageing field. There are some limitations that have been appropriately included within the discussion section; however, I would suggest the authors to address the following minor comments:

  • Figure 2: For myotube identification, it would have been more precise to use a market for the identification and quantification of differentiated muscle cells, such as MF 20 staining. Also, it is difficult to visualise the myotubes. Please could the authors include figures with a higher resolution, at least 300dpi?
  • The complete RNA-sequencing data should be included as supplementary data or in a website repository. A heatmap of the DEGs would be also encouraged for illustrative purposes.
  • Table 1: please, include the FDR values of each gene included in table 1.
  • If possible, it would be ideal to include individual points in the graphs.
  • Line 74: ‘(…) did not affect the mRNA expression levels of either myogenin or CK protein’. The authors need to include a brief description (one sentence) of the reasons looking at myogenin mRNA and CK protein in the muscle. Why was only the myogenin mRNA analysed whereas only the CK protein was included?
  • Figure 4A: it would be better if the authors include a representative image of each condition with a lower magnification showing more myotubes per image field.
  • Figure 4C: although this is a supporting result of the previous figure, it would have been more reliable if the authors performed this experiment using a more conclusive approach for the identification of apoptotic cells (e.g. acridine orange/ethidium bromide or any other commercialised viability assay).
  • Line 240: it would be more appropriate to change the title to a general description of the corresponding section.
  • Lines 288-289: it seems that this sentence does not belongs here. Please, remove if appropriate.
  • Can the decrease in the total protein content be due to an increase in apoptosis? Please, search for literature in this respect and, if relevant, include a possible hypothesis to explain this finding.

Round 2

Reviewer 1 Report

Are siScramble data the same in Figures 1B and C?

A second siRNA should be used to confirm some of the findings. This is common practice in molecular and cellular biology.

When the authors write N = 6 it is unclear what this means. Were 6 independent plates of cells treated with the siRNA? And then for each experiment, the sample was run in duplicate?

What is the biological significance of the reported decreased protein content in Figure 1E? The decrease in protein did not impact the differentiation of the cells based on the number of myonuclei per myotube or on the size of the myotube. It is very difficult to make out one myotube from another in Figure 2A. I wonder how the authors determined this.

The notion that KD of MBNL1 leads to greater apoptosis is weak and not convincing. In fact, as far as I can tell, nothing in the RNAseq data is suggestive of ongoing apoptosis.

Regarding the authors comment for remark #5, the authors will want to include multiple reference supporting their claims that muscles from 2-month-old mice are still developing.

Round 3

Reviewer 1 Report

At the very least, if the authors can't conceptually explain the protein content data they should seriously consider removing it from the study. It adds nothing of substance to study and it would prevent the authors from making statements like the following: 

"However, we have no clear explanation regarding this issue at present.”